# Dry-Aged Beef Steaks: Effect of Dietary Supplementation with *Pinus taeda* Hydrolyzed Lignin on Sensory Profile, Colorimetric and Oxidative Stability

**DOI:** 10.3390/foods10051080

**Published:** 2021-05-13

**Authors:** Aristide Maggiolino, Maria Federica Sgarro, Giuseppe Natrella, Josè Manuel Lorenzo, Annamaria Colucci, Michele Faccia, Pasquale De Palo

**Affiliations:** 1Department of Veterinary Medicine, University of Bari A. Moro, Valenzano, 70010 Bari, Italy; maria.sgarro@uniba.it (M.F.S.); pasquale.depalo@uniba.it (P.D.P.); 2Department of Soil, Plant and Food Sciences, University of Bari Aldo Moro, Via Amendola 165/A, 70126 Bari, Italy; giuseppe.natrella@uniba.it (G.N.); annamaria.colucci@uniba.it (A.C.); michele.faccia@uniba.it (M.F.); 3Centro Tecnológico de la Carne de Galicia, Rúa Galicia 4, Parque Tecnológico de Galicia, San Cibrán das Viñas, 32900 Ourense, Spain; jmlorenzo@ceteca.net; 4Area Tecnologia de los Alimentos, Facultad Ciencias de Oruesnse, Universidad de Vigo, 32004 Ourense, Spain

**Keywords:** beef, meat aging, volatile compounds, sensory profile, oxidative profile, polyphenols

## Abstract

Flavor is one of the main factors involved in consumer meat-purchasing decision and use of natural antioxidants in animal feeding had a great appeal for consumers. The aim of this trial is to evaluate the effect of *Pinus taeda* hydrolyzed lignin (PTHL) feed addition on oxidative stability, volatile compounds characteristics, and sensory attributes of 35 days dry-aged beef steaks. Forty steer six months old were randomly divided into a control group (CON; *n* = 20) and an experimental group (PTHL; *n* = 20). Both groups were fed ad libitum for 120 days with the same TMR and only the PTHL group received PTHL supplement. Samples of LT muscle were removed from carcasses and dry aged for 35 days at 2 °C, 82% of humidity, and 0.4 m/s of ventilation and then analyzed. Meat of CON group showed lower yellowness (*p* < 0.01) and higher TBARS (*p* < 0.01) values. Moreover, CON meat showed higher volatile aldehydes and lower sulfur compounds (*p* < 0.01), with higher unpleasant odor (*p* < 0.05) and meaty odor (*p* < 0.01) score revealed by sensory assessors. PTHL inclusion in beef diet delayed the oxidative mechanisms in 35 days dry-aged steaks, resulting in an improved colorimetric, volatolomic, and sensory profile.

## 1. Introduction

Beef is an important source of high biological value proteins, vitamins, and micronutrients for human nutrition. Consumer acceptability is affected by some aspect of meat quality like tenderness, color, juiciness, and cooking meat flavor and aroma [1,2]. In particular, consumers tend to prefer tender meat with natural taste and aroma [3]. Flavor is one of the main factors involved in the purchasing decision [4], representing the food sensory impression of taste and smells combined, and is also influenced by other factors able to affect the individual flavor perception, such as tenderness and juiciness [5].

In past years, the interest of consumers toward dry-aged steaks has strongly increased due to their flavor that arises during aging [6]. The flavor and taste precursors of meat products derive from the thermal transformation of non-volatile compounds in raw meat, like amino acids, peptides, reducing sugars, vitamins, and nucleotides [3]. On this side, many factors as breed, sex, age, animal feeding, meat ageing, and cooking method can influence taste and aroma [7,8]. Moreover, fatty acid composition, intramuscular fat content, and oxidative stability influence the texture, juiciness, taste, and consequently flavor [9], as well as meat shelf-life [10].

With a view to a circular economy, the inclusion of agricultural by-products in ruminant feeding could have an important environmental and social impact [11], furthermore, the high presence of bioactive compounds in vegetal matrices makes by-products very interesting for the improvement of the nutritional properties, as well as of the chemical-physical and sensory characteristics of beef. Plants are widely considered as the main source of natural antioxidant like polyphenolic compounds (flavonoids, tannins, and phenolic acids) [12]. Their presence in meat can delay the lipids and proteins oxidation, thus protecting the product from deterioration [13]. Many experimental trials focused on the use of some antioxidants in ruminants’ nutrition and their effects on meat quality, fatty acids profile, and oxidative stability [14,15,16,17]. In general, antioxidant diet addition is considered a good strategy for delaying the lipid oxidation in meat, improving its shelf-life, but contradictory results have been often reported. Conflicting results could be due to the different types of antioxidants used (i.e., variability of vegetal origin and bioactive compounds) and/or to different aging conditions investigated. One of these antioxidants is *Pinus taeda* hydrolyzed lignin (PTHL), derived from *Pinus taeda*, known as loblolly pine, a very common species of tree in the Northern America. It is a mixture of polyphenols obtained by the hydrolyzation of lignin from the above mentioned tree species, that proved to have positive effect on animal welfare and gas emission [18] and on beef meat slices quality during 14 aging days [14]. The present study deals with the influence of this antioxidant on meat quality over a longer aging time. In particular, it aimed to determine color stability, oxidative status volatile compounds, and sensory characteristics of steaks from dry-aged beef loins over 35 days of retail display.

## 2. Materials and Methods

### 2.1. Animals and Sampling

The research for animal testing was approved by the Ethics Committee for animal testing (CESA) (process number 2-X/17). Forty Limousine steers six months old were included in the trial. The experimental design was the same described by Maggiolino, Lorenzo, Quiñones, Latorre, Blando, Centoducati, Dahl and De Palo [18]. All beefs were randomly assigned to two groups of 20 animals each. Each group was in a separated pen, equipped with automatic waterers. One was the experimental group (PTHL) and the other one the control group (CON). All animals were ad libitum fed with the same total mix ration (TMR) (Table 1) for 120 days, until slaughtering procedures at 10 months of age.

The PTHL group received the *Pinus taeda* hydrolyzed lignin additive (PTHL, Oxyphenol^®^, I-Green, Padua, Italy).

The chemical composition of *Pinus taeda* hydrolyzed lignin and its antioxidant activity were calculated [19,20] and are reported in Table 2.

The PTHL group received 35 g/day beef each during the 90 days of the trial and 70 g/day beef each for the last 30 days [14]. It was orally administered to each beef in the self-locking head gate in the feeding front when TMR was unloaded, as described by Maggiolino, Lorenzo, Salzano, Faccia, Blando, Serrano, Latorre, Quiñones and De Palo [14].

After slaughtering (1099/2009EC; EC 2009), carcasses were chilled at 4 °C for 24 h. Subsequently, samples of LT muscle from the 13th and the 15th thoracic vertebra were removed from the left half carcass of each animal and dry aged for 35 days in an aging room at 2 °C, 82% of humidity, and 0.4 m/s of ventilation. Then the aging meat was analyzed.

### 2.2. Colorimetric Analysis

Before color measurements, muscles were freshly cut and allowed to bloom directly in contact with air for 20 min. Colorimetric parameters (L*, a*, and b*) were measured as described by Gálvez et al. [21], using a Minolta CR-300 colorimeter (Light Source D65, Minolta Camera Co., Osaka, Japan). Measurements were collected from a 0° viewing angle with A-pulsed xenon arc lamp, with a reading surface of 8-mm diameter. They were performed in three different points of each sample and, on each point, three measurements were performed rotating the detector system by 90° compared to the previous one, for a total of nine measurements for each sample. The mean values were considered for statistical analysis. Chroma = (a^2^ + b^2^)^1/2^ and hue (°) = tan^−1^ (b/a) were calculated according to De Palo et al. [22].

### 2.3. Water Holding Capacity (WHC), Warner Blatzer Shear Force (WBSF), Cooking Loss, and pH

Water-holding capacity was determined as described by De Palo et al. [23] using the centrifugation method. A meat sample of 0.3 g was centrifuged at 30,000× *g* for 1 h. After centrifugation, the samples were dried and weighed again, and the centrifugation loss was calculated as the difference in weight before and after centrifugation.

The Warner–Braztler Shear Force was performed following procedure described by Gálvez, Maggiolino, Domínguez, Pateiro, Gil, De Palo, Carballo, Franco and Lorenzo [21]. Meat samples were cooked placing vacuum package bags in a water bath with automatic temperature control (JP Selecta, Precisdg, Barcelona, Spain) at 85 °C to an internal temperature of 75 °C, measured with a copper constantin fine-wire thermocouple fixed in the geometrical center of the sample (Model 5SCTT-T-30-36; Omega Engineering Inc., Norwalk, CT, USA). Three pieces of meat of 1 × 1 × 4 cm (height × width × length) were removed parallel to the muscle-fiber direction. Samples were completely cut using a WB shear blade with a triangular slot-cutting edge (1-mm thickness) at a cut speed of 20 cm/min and a load of 50 N using an Instron 1140 apparatus (Instron, High Wycombe, UK). After cooking, same samples were cooled and the percentage of cooking loss was calculated as described by De Palo et al. [24].

For pH recording a portable pH meter with shaped glass electrode was used to easily penetrate meat (Carlo Erba pH 710; Carlo Erba Reagenti, Milano, Italy). It was automatically calibrated for muscle temperature and using solutions with 4 and 7 pH values (Crison, Lainate, Italy) [25].

### 2.4. Thiobarbituric Acid Reactive Substances (TBARS), Hydroperoxides, and Protein Carbonyls Analyses

For TBARS and hydroperoxides analysis, 5 g and 2 g respectively of raw meat were previously minced and placed in a 50-mL tube and then homogenized with 15 mL deionized distilled water (DDW). The TBARS were determined as reported by Tateo et al. [26].

Hydroperoxides determination was performed according to De Palo et al. [27].

Protein carbonyls determination was performed as reported by De Palo et al. [28].

Briefly, a 2 g meat sample was previously homogenized in 20 mL of 0.15 M KCl for 2 min. Two aliquots of homogenate (50 µL each) were added with 1 mL 10% TCA and then centrifuged at 1200× *g* for 3 min at 4 °C to measure protein oxidation. One aliquot was used as standard adding 1 mL of 2 M HCl solution. The second one was added with 1 mL of 2 M HCl containing 10 mM 2,4-dinitrophenyl hydrazine (DNPH).

### 2.5. Superoxide Dismutase, Catalase and Glutathione Peroxidase Activity Evaluation

Samples (2 of 400 mg) of raw meat were homogenized in a tissue homogenizer 4 mL saline at 4 °C. After centrifugation at 4 °C for 20 min at 7000× *g*, the supernatant was separated and analyzed for the antioxidant enzyme activity determination. Superoxide dismutase (SOD, EC 1.15.1.1), catalase (CAT, EC 1.11.1.6), and glutathione peroxidase (GPx, EC1.11.1.9.) activity was measured as described by Maggiolino et al. [29]. The SOD activity was determined from its ability to inhibit the epinephrine autoxidation. One SOD unit is defined as the enzyme quantity necessary to inhibit the rate of epinephrine autoxidation by 50% [30]. The CAT was measured according to the absorbance decreasing of H_2_O_2_ at 240 nm (e = 40 M^−1^ cm^−1^). One unit of enzyme activity is defined as the amount of enzyme that is required to degrade 1 micromole of H_2_O_2_ in 1 min. The GPx activity was measured following the GSH oxidation rate by *tert*-butyl hydroperoxide, catalyzed by GPx [31].

### 2.6. Volatile Compounds (VOC) Analysis

Five grams meat samples were (Delonghi, Mod. CG660, Treviso, Italy) at 130–150 °C until 70 °C at core was reached. Temperature was measured with a copper constantin fine-wire thermocouple fixed in the geometrical center of the sample (Model 5SCTT-T-30-36; Omega Engineering Inc., Norwalk, CT, USA) as described by Maggiolino et al. [32]. Then, samples were minced using a commercial grinder (Moulinex/Swan Holding Ltd., Birmingham, UK), and analyzed. The volatile compounds were extracted by solid-phase micro-extraction (SPME) as described by Natrella et al. [33]. The samples were weighed (1 ± 0.05 g) into 20 mL vials, closed by a rubber septum and an aluminum cap. All samples were added with internal standard (82 ng 2-octanol) to perform a semi-quantitation and loaded into an autosampler Triplus RSH (Thermo Fisher Scientific, Rodano, Italy).

### 2.7. Sensory Analysis

The sensory analysis test was performed by a panel composed of twelve trained assessors. They were selected according to their sensory acuity according to the British Standards Institution (BSI, 1993) methods. The meat samples for sensory analysis were cut into slices (about 2 cm thin) and cooked as described above for VOCs determination. After removing fat and connective tissue, the muscle was cut into about 2 cm^3^ pieces, then wrapped in pre-labelled foils and placed in a heated incubator until offered to the assessors. The samples tasting order designs were based on what was reported by MacFie et al. [34] aiming to balance the carryover effects between samples.

The panel test was organized in eight different sitting sessions for each panelist, in which five samples were tasted. In total each panelist received 40 samples (two samples for each of the 20 beef steers). A sensory panel software was used to randomize the samples order presentation for each assessor. Tested samples were scored on a 1–10 point scale for tenderness (1 = extremely tough to 10 = extremely tender), juiciness (1 = extremely dry to 10 = extremely juicy), overall assessment (1 = extremely disliking to 10 = extremely licking), sweetness, unpleasant taste, meaty odor, and unpleasant odor, (1 = extremely weak to 10 = extremely strong).

### 2.8. Statistical Analysis

The data set was tested for normal distribution and variance homogeneity (Shapiro-Wilk). Each beef muscle represented an experimental unit. The data set was subjected to analysis of variance (ANOVA) using the GLM by SAS software [35], according to the following model:y_i_ = μ + α_i_ + T_i_ + ε_ij,_
where y_ij_ represents all the previous cited patterns as dependent variables; μ is the overall mean; α_i_ is the constant of the beef random effect; T was the effect of the ith inclusion of the *Pinus taeda* hydrolyzed lignin in the diet (i = 1, 2) and ε_ij_ was the error term. The significance was set at *p* < 0.05, and the results were expressed as means and mean standard error.

## 3. Results

The effects of *Pinus taeda* hydrolyzed lignin diet addition on colorimetric parameters, tenderness, and meat properties are reported in Table 3.

Yellowness and Hue angle presented higher (*p* < 0.01) mean values in PTHL meat than CON group. On the contrary, CON meat showed higher lightness values (*p* < 0.01). Moreover, the higher WHC was reported for PTHL meat (*p* < 0.05). However, no differences were observed for redness, WBSF, cooking loss, and pH (*p* > 0.05).

Table 4 shows oxidative parameters and enzyme activities in meat from animals of the PTHL and CON groups.

TBARS showed higher (*p* < 0.01) values in CON group, whereas enzyme activity of glutathione peroxidase resulted higher (*p* < 0.01) in PTHL meat. Hydroperoxides and protein carbonyls concentrations, as well as SOD and CAT activities did not show differences between experimental groups (*p* > 0.05).

The effect of the PTHL dietary supplementation on the VOC profile is shown in Table 5, Table 6, Table 7 and Table 8.

Table 5 describes the VOC groups. Hydrocarbons, aromatic hydrocarbons, and aldehydes have higher concentrations in meat from CON group (*p* < 0.01), that showed lower sulfur compounds (*p* < 0.01). Particularly, aromatic hydrocarbons and carboxylic acids indicated that decane, 2,2 dimethyl resulted lower in amounts in CON meat (*p* < 0.01), instead eicosane, 2-methyl-, nonadecane 2-methyl (*p* < 0.01), and octane (*p* < 0.05) had higher concentrations (Table 6). The aromatic hydrocarbon m-xylene showed lowest values in PTHL (*p* < 0.01). Moreover, PTHL meat showed lower values of butanoic (*p* < 0.05), heptanoic (*p* < 0.01), octanoic (*p* < 0.01), and nonanoic acid (*p* < 0.01) compared to CON. Volatile aldehydes and ketones are reported in Table 7. Among aldehydes group, butanal 2-methyl, benzaldehyde 3-ethyl (*p* < 0.05), butanal, 3-methyl-, pentanal, hexanal, heptanal, heptanal 2-methyl, nonanal (*p* < 0.01) recorded higher concentrations in CON group. With regard to ketones, 2,3 pentanedione (*p* < 0.01) and 2-butanone (*p* < 0.05) had lower amounts in meat from PTHL group, that also showed greater 1-hepten-3-one values (*p* < 0.01). Table 8 shows the effect of dietary supplementation with PTHL in volatile alcohols, furans, sulfur compounds, and pyrazines. As to alcohols, 2-propanol (*p* < 0.01) and 1-hexanol (*p* < 0.05) resulted higher in meat from CON group, disulphide dimethyl (*p* < 0.01) resulted greater in PTHL meat.

Results of sensory evaluation of meat from both the experimental groups are reported in Table 9.

## 4. Discussion

Color is the main attribute that influences consumer purchase and acceptability of meat [36] and it has become an important factor for the meat industry and manufacturer to pay attention to [37]. Moreover, it is widely known to be affected by aging, but the mechanisms by which it is affected under retail display conditions are overall unknown [6]. Our results showed that *Pinus taeda* hydrolyzed lignin supplementation affects colorimetric parameters of 35-days dry-aged beef. Although redness and chroma were not affected, the antioxidant addition led to lower lightness values and higher yellowness and hue values after 35 dry-aging days. The lower lightness could be explained by less light reflection and this should be linked to the moisture loss [6,38], and to the different water-holding capacity. During aging there is a muscular fibers breakdown that can let the water pass from the intracellular to the extracellular districts, influencing the meat reflectance and consequently lightness and water-holding capacity [39]. It could be supposed that this happened with lower intensity in meat of animals fed with antioxidant addition. In fact, meat from these animals was characterized by higher water-holding capacity. Previous studies [14], although conducted on meat slices and not on entire cuts (as in the present trial is) and with a shortest aging time (15 days), reported no effect of antioxidants dietary supplementation on meat lightness nor redness, but a great effect on yellowness. Similarly, considering these two last parameters, our results reported that at 35 days, on a fresh cutting surface, yellowness values resulted higher in meat of animals fed with *Pinus taeda* hydrolyzed lignin. Red meat is characterized by different oxidation processes that involve multiple substrates and affect colorimetric parameters. Haem pigment, the most important compound affecting meat redness [40], is believed to be linked also to lipid peroxidation, serving as a catalyst to these reactions [41]. Lipid oxidation itself is one of the most important processes affecting meat yellowness. Probably, the antioxidant activity exerted by the polyphenols characterizing the *Pinus taeda* hydrolyzed lignin was able to reduce meat oxidation, decreasing the yellowness after aging [42]. This is an important result from both the consumers’ and sellers’ point of view, in fact researchers reported that b* values are positively correlated with consumer meat appreciation whereas redness is negatively correlated to it [43].

The pH values did not showed differences due to polyphenols addition, and the values observed were those usually reported in aged beef meat [44]. It is well known that the ultimate pH can affect meat quality, particularly rheological parameters such as water-holding capacity and cooking losses [45], but the lack of differences and the optimal values reached after 35 days could be related to the lack of differences in cooking loss, although WHC was slightly affected. Tenderness was not affected by polyphenols addition, since our values can be considered good ones for 35 days-aged beef steaks [46].

The *Pinus taeda* dietary supplementation is associated with the production of meat characterized by lower TBARS levels, similarly to what is reported by other authors [42], but no differences were observed for other oxidative catabolites. It was reported that dietary polyphenols inclusion in ruminants can affect fatty acid metabolism and its metabolic pathways, influencing the lipid oxidation [14,45]. This result can be related to the higher glutathione peroxidase activity observed. This enzyme is one of the in vivo cell defense systems against oxidative damage [47] and polyphenols dietary addition seems to enhance it, probably improving some anti-oxidative defense equipment [42] resulting also in best meat color stability and a reduced oxidative catabolites production [14].

Raw meat is weakly flavored, as the heat process during cooking provides a wide number of volatile compounds and intermediate products, produced through Maillard reaction, lipid oxidation, and/or vitamin degradation [3,7], enhancing meat flavor development and aroma generation. Our trial resulted in a total of 55 volatile compounds, identified after the grill cooking process. They are classified as hydrocarbons (*n* = 7), aromatic hydrocarbons (*n* = 3), carboxylic acids (*n* = 6), aldehydes (*n* = 11), ketones (*n* = 12), alcohols (*n* = 12), furans (*n* = 1), sulfur compounds (*n* = 2), and pyrazines (*n* = 1).

As reported by current literature in beef, lamb, pork, and equids meat, aldehydes are the predominant chemical group after cooking process and hexanal is the most abundant [8,26,29,32,48,49,50,51,52]. Hexanal is responsible of the meat meaty, grassy, and fatty odors, and is positively correlated with consumers’ sensory evaluation. Meat derived from animals fed with PTHL addition is characterized by lower aldehydes release, and also by lower concentration of hydrocarbons and aromatic hydrocarbons. These last VOCs families have less impact on meat flavor, probably also due to their relatively high odor threshold values [53]. The lower volatile aldehydes content can be due to the reduction of the rate of lipid and protein oxidation and, consequently, the decreasing of production of precursor, main end products, and chemical group composition of volatile compounds [11,54,55]. Moreover, pentanal, that is associated to rancid off-flavor (metallic, green, earthy, beany) [56], is less produced from meat from the PTHL groups. This could potentially have a good impact on the sensory evaluation, reducing the off-flavors, considering the low flavor thresholds that characterized aldehydes.

Butanoic acid and other carboxylic acids, as ethanoic acid, can arise from amino acids fermentation by Strickland reaction [9]. However, although these acids are released in lower quantities by PTHL meat, they are generally poorly produced. In the same way, sulfur compounds have low concentrations, although they are higher in samples from the PTHL group. Some authors reported that these compounds were found to be desirable in relation to cooked beef flavor [57]. The sulfur-containing compounds should be produced by the degradation of sulfur amino acids, such as cysteine and methionine [7]. Alcohols can be secondary products of lipid oxidation, and among these, 2-propanol and 1-hexanol were affected by dietary PTHL supplementation, showing lower values. These are considered responsible for resin, flower and green aroma [58], and the lower presence of these in PTHL meat could indicate a decrease in lipid oxidation with a potential positive effect on sensory evaluation.

*Pinus taeda* hydrolyzed lignin affected some sensory patterns. Both samples from PTHL and CON groups, at 35 days of ageing, had an overall positive evaluation by assessors. Meat from animals fed with polyphenols addition resulted juicier, probably due to differences in WHC that, although minimal, might have been able to affect assessors’ juicy perception. Moreover, the differences observed in the VOCs amount release may have impacted the meat aroma evaluation. Aldehydes had great impact on sensory evaluation [59], and differences in hexanal production, the most abundant VOC, characterized by relatively low perception thresholds and perceived as meaty odor, should be responsible of the differences in meaty odor evaluation. Although meaty odors showed higher sensory test scores in the meat from CON group, it must be underlined that this meat was scored with higher unpleasant odor scores. Probably, the higher pentanal production can also have affected this sensory evaluation because of its high correlation with rancid off-flavor [56]. Despite these contrasting results, meat obtained with PTHL dietary supplementation resulted in a better overall assessment score.

## 5. Conclusions

The inclusion of *Pinus taeda* in beef diet delayed the oxidative pathways in 35 days dry-aged steaks. Particularly, an increase of glutathione peroxidase activity was observed in the PHTL meat, which, consequently, resulted in a better oxidative stability and a positive increase of yellowness. Moreover, the addition of PTHL improved the profile of volatile compounds and sensory panel evaluation. Although hexanal production is inhibited, also off-flavor volatile compounds generation as pentanal is reduced. Furthermore, beef obtained by animals supplemented with PTHL resulted in better juiciness and overall assessment score. Our results showed that use of natural substances with high antioxidant activity can be considered an effective tool for the improvement of meat quality and of oxidative stability.

## Figures and Tables

**Table 1 foods-10-01080-t001:** TMR ingredients and nutrient composition (g/kg on dry matter basis) fed to all animals for 120 days.

Ingredients	
Wheat straw	150
Ground maize	440
Soybean meal solvent 44% protein	140
Ground barley	125
Wheat bran	110
Hydrogenated triacylglyceride from palm oil	10
Mineral/vitamin supplement	25
**Nutrient Composition**	
Dry matter	867.3
Organic matter	852.2
Crude protein	159.2
Crude fiber	92.2
NDF	256.1
ADF	101.3
ADL	26.3
Ether extract	39.4
Ash	40.1

NDF, neutral-detergent fiber; ADF, acid-detergent fiber; ADL, acid-detergent lignin.

**Table 2 foods-10-01080-t002:** Composition and antioxidant activity of *Pinus taeda* hydrolyzed lignin (Oxyphenol^®^) fed to beef steers for 120 days.

Components (g/kg)	
Vanillin	264
Eriodictyol	34
Quercetin	27
Isorhamnetin	16
Rosmarinic acid	14
Quercetin rhamnoside	139
Methyl gallate rutinoside	423
Epigallocatechin-3-methylgallate	15
Ferulic acid derivatives	67
Antioxidant activity (µmol TE g^−1^ DW)	
TEAC	23.9
ORAC	122.44

TE, trolox equivalents; DW, dry weight; TEAC, trolox equivalent antioxidant capacity; ORAC, oxygen radical absorbance capacity.

**Table 3 foods-10-01080-t003:** Effect of *Pinus taeda* hydrolyzed lignin diet addition on colorimetric and rheological parameters of beef meat dry aged for 35 days (*n* = 20 samples for each experimental group).

	PTHL	CON	SEM	*p*-Value
**Colorimetric Parameters**				
Lightness (L*)	40.58	44.02	0.65	0.0044
Redness (a*)	12.44	11.05	0.43	0.4843
Yellowness (b*)	1.42	−2.48	0.14	<0.0001
Chroma	10.88	10.54	0.55	0.3594
Hue angle (radians)	0.07	−0.12	0.01	0.0004
**Tenderness and Meat Properties**				
WBSF (N)	44.56	42.15	1.23	0.5821
Cooking Loss (%)	38.45	40.15	3.52	0.3992
WHC (%)	81.41	74.33	2.05	0.0444
pH	5.57	5.61	0.02	0.6124

PTHL, *Pinus taeda* hydrolyzed lignin group; CON, control group; SEM, standard error of the mean; WBSF, warner Blatzer Shear Force; WHC, water holding capacity.

**Table 4 foods-10-01080-t004:** Effect of *Pinus taeda* hydrolyzed lignin diet addition on oxidative parameters and enzyme activity of beef meat dry aged for 35 days (*n* = 20 samples for each experimental group).

	PTHL	CON	SEM	*p*-Value
**Oxidative Parameters**				
TBARS	0.48	0.71	0.02	0.0005
Hydroperoxides	0.32	0.33	0.01	0.4788
Protein Carbonyls	2.61	6.53	0.04	0.2881
**Enzyme Activity**				
Superoxide dismutase (U/mg protein)	35.32	37.43	2.02	0.4662
Catalase (U/mg protein)	3.93	4.05	0.08	0.3145
Glutathione peroxidase (µmol NADPH ox/mg protein)	0.13	0.09	0.008	0.0010

PTHL, *Pinus taeda* hydrolyzed lignin group; CON, control group; SEM, standard error of the mean.

**Table 5 foods-10-01080-t005:** Effect of *Pinus taeda* hydrolyzed diet addition on total hydrocarbons, aromatic hydrocarbons, carboxylic acids, aldehydes, ketones, alcohols, furans, sulfur compounds, and pyrazines of beef meat dry aged for 35 days (*n* = 20 samples for each experimental group). Results are expressed as ng/g of meat.

Volatile Compound (Total)	PTHL	CON	SEM	*p*-Value
Hydrocarbons	449.74	724.41	20.90	<0.0001
Aromatic hydrocarbons	62.70	116.13	7.23	<0.0001
Carboxylic acids	210.13	300.40	30.90	0.0555
Aldehydes	2095.51	4646.21	324.68	<0.0001
Ketones	1021.62	1347.09	130.45	0.0968
Alcohols	404.78	448.40	38.36	0.4331
Furans	6.03	7.96	2.14	0.5336
Sulfur compounds	22.35	10.55	1.56	<0.0001
Pyrazines	11.34	11.95	0.66	0.5224

PTHL, *Pinus taeda* hydrolyzed lignin group; CON, control group; SEM, standard error of the mean.

**Table 6 foods-10-01080-t006:** Effect of *Pinus taeda* hydrolyzed lignin diet addition on hydrocarbons, aromatic hydrocarbons, and carboxylic acids of beef meat dry aged for 35 days (*n* = 20 samples for each experimental group). Results are expressed as ng/g of meat.

Volatile Compound	PTHL	CON	SEM	*p*-Value
**Hydrocarbons**				
heptane, 2,2,4,6,6-pentamethyl	33.65	26.63	4.78	0.3149
hexane, 3,3,4,4-tetrafluoro-	13.76	16.99	2.09	0.2913
2,2,7,7-tetramethyloctane	61.86	56.32	6.33	0.5445
Octane	25.20	49.15	7.82	0.0459
nonadecane 2-methyl	104.91	223.51	20.27	0.0008
eicosane, 2-methyl-	133.60	312.25	14.49	<0.0001
decane, 2,2 dimethyl	76.76	39.56	3.55	<0.0001
**Aromatic Hydrocarbons**				
*o*-xylene(benzene, 1,2-dimethyl-)	25.83	42.84	6.64	0.0888
*m*-xylene(benzene, 1,3-dimethyl-)	19.61	56.96	1.95	<0.0001
ethylbenzene	17.25	16.32	1.41	0.6463
**Carboxylic Acids**				
hexanoic acid	51.50	27.35	8.46	0.0657
butanoic acid	42.32	76.38	11.01	0.0439
heptanoic acid	7.65	25.03	8.64	0.0005
octanoic acid	10.99	38.33	4.19	0.0003
nonanoic acid	29.22	63.31	6.94	0.0031
ethanoic acid	68.45	69.99	23.22	0.9631

PTHL, *Pinus taeda* hydrolyzed lignin group; CON, control group; SEM, standard error of the mean.

**Table 7 foods-10-01080-t007:** Effect of *Pinus taeda* hydrolyzed lignin diet addition on aldehydes and ketones of beef meat dry aged for 35 days (*n* = 20 samples for each experimental group). Results are expressed as ng/g of meat.

Volatile Compound	PTHL	CON	SEM	*p*-Value
**Aldehydes**				
butanal, 2-methyl- (butyraldehyde, 2-methyl-)	29.35	52.77	5.82	0.0117
butanal, 3-methyl- (isovaleraldehyde)	20.80	45.52	3.93	0.0004
pentanal	95.95	236.64	21.67	0.0003
hexanal (*n*-caproaldehyde)	1709.88	3924.47	309.94	0.0001
heptanal	42.48	120.66	12.16	0.0003
heptanal, 2-methyl-	30.11	78.05	4.45	<0.0001
benzaldehyde	41.32	43.27	4.29	0.7516
Benzaldehyde, 3-ethyl-	8.08	12.16	1.04	0.0138
nonanal	48.48	70.56	3.77	0.0008
octanal	59.04	53.73	4.94	0.4586
Acetaldehyde	10.02	8.37	1.70	0.5051
**Ketones**				
2-octanone	113.07	90.46	7.79	0.0568
5-hepten-2-one, 6-methyl-	14.64	13.60	2.57	0.7795
2,3 butanedione (biacetyl)	193.06	277.50	47.24	0.2244
acetone (2-propanone)	113.19	114.78	24.08	0.9634
1-hepten-3-one	10.38	4.54	0.78	<0.0001
3-nonanone	2.60	3.40	0.42	0.2030
2-heptanone	25.20	20.09	6.26	0.5720
acetol (2-propanone, 1-hydroxy)	3.57	3.26	0.46	0.6440
3-acetyl-1*H*-pyrroline	5.21	4.94	0.65	0.7668
2,3 - pentanedione	4.75	16.93	0.83	<0.0001
2-butanone	38.94	75.12	9.09	0.0125
acetoin (2-butanone,3-hydroxy-)	503.18	729.13	90.91	0.0979

PTHL, *Pinus taeda* hydrolyzed lignin group; CON, control group; SEM, standard error of the mean.

**Table 8 foods-10-01080-t008:** Effect of *Pinus taeda* hydrolyzed diet addition on alcohols, furans, sulfur compounds, and pyrazines of beef meat dry aged for 35 days (*n* = 20 samples for each experimental group). Results are expressed as ng/g of meat.

Volatile Compound	PTHL	CON	SEM	*p*-Value
**Alcohols**				
1-butanol, 3-methyl-	53.13	43.14	6.76	0.3116
1-pentanol	81.93	88.44	18.21	0.8037
2-heptanol, 2,6-dimethyl-	13.92	17.04	1.75	0.2243
2-propanol	45.12	46.14	7.34	0.9231
2-propyl-1-pentanol	19.09	19.07	2.75	0.9951
2-propanol, 1-(2-methoxypropoxy)	40.50	74.45	4.65	<0.0001
ethanol	34.87	47.72	4.35	0.0531
1-hexanol, 2-ethyl-	43.66	57.86	4.56	0.0428
carbitol	3.21	3.61	0.47	0.5587
1-penten-3-ol	5.35	5.08	0.52	0.7157
1-octen-3-ol (amyl vinyl carbinol)	50.99	30.63	11.89	0.2434
eucalyptol	13.00	15.22	2.65	0.5630
**Furans**				
furan, 2-pentyl-	6.03	7.96	2.14	0.5336
**Sulfur Compounds**				
dimethyl sulfoxide	2.81	1.92	0.46	0.1868
disulfide, dimethyl	19.54	8.63	1.57	0.0002
**Pyrazines**				
methylpyrazine	3.30	3.70	0.31	0.3727

PTHL, *Pinus taeda* hydrolyzed lignin group; CON, control group; SEM, standard error of the mean.

**Table 9 foods-10-01080-t009:** Effect of *pinus taeda* hydrolyzed diet addition on sensory evaluation of beef meat dry aged for 35 days (*n* = 20 samples for each experimental group).

Sensory Profile	PTHL	CON	SEM	*p*-Value
Tenderness	7.78	7.55	0.06	0.4478
Juiciness	7.75	7.04	0.03	0.0392
Sweetness	6.52	6.67	0.04	0.2122
Unpleasant taste	3.35	3.65	0.06	0.1558
Unpleasant odor	4.45	5.12	0.04	0.0281
Meaty odor	6.68	7.54	0.05	0.0015
Overall assessment	7.75	7.05	0.06	0.0043

PTHL, *Pinus taeda* hydrolyzed lignin group; CON, control group; SEM, standard error of the mean.

## Data Availability

Not applicable

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
