# Peer review of "Dry-Aged Beef Steaks: Effect of Dietary Supplementation with Pinus taeda Hydrolyzed Lignin on Sensory Profile, Colorimetric and Oxidative Stability"

_foods, 2021, doi:10.3390/foods10051080_

Round 1

Reviewer 1 Report

The manuscript contains interesting information on the effect of dietary supplementation with Pinus taeda hydrolyzed lignin on sensory profile, colorimetric and oxidative stability in dry aged beef steaks. The results of this study are well documented by tables. The discussion of the data does not inspire any doubts. The literature references are recent and adequate.

 Table 3 and line 258 – in my opinion cooking loss, WHC, pH are not rehological parameters, but these are the characteristics/properties of meat

Line 28 Aldehydes, Sulphur use lowercase letters instead of uppercase letters

Line 42, 57, 58, 366 - no spaces before bracket with citing literatureTable 9 - overall acceptability is used in consumer sensory evaluation, there should be an

overall assessment or score or only overall, because the sensory analysis was performed by a panel of trained assessors.

References should be described as follows, depending on the type of work:

  • Journal Articles:
    1. Author 1, A.B.; Author 2, C.D. Title of the article. Abbreviated Journal Name Year, Volume, page range.

Check all references: Line 455, 482, 504, 522, 529, 533, 535, 539, 541, 556, 571, 594 - no journal name

Meat Sci. instead Meat science

Line 478 - check Chemical–nutritional…

Line 488 exist “beef %J Animal Production Science”

Line 509 exist “Organic %J Annals of Animal Science”

Line 524 exist “life1. Journal…”

Author Response

Reviewer 1

The manuscript contains interesting information on the effect of dietary supplementation with Pinus taeda hydrolyzed lignin on sensory profile, colorimetric and oxidative stability in dry aged beef steaks. The results of this study are well documented by tables. The discussion of the data does not inspire any doubts. The literature references are recent and adequate.

AU: dear reviewer, thank you very much for your comments about this research paper. We are grateful.

Table 3 and line 258 – in my opinion cooking loss, WHC, pH are not rehological parameters, but these are the characteristics/properties of meat

AU: thank you for your suggestion. We modify both table 3 and line 258

Line 28 Aldehydes, Sulphur use lowercase letters instead of uppercase letters

AU: done

Line 42, 57, 58, 366 - no spaces before bracket with citing literature. Table 9 - overall acceptability is used in consumer sensory evaluation, there should be an overall assessment or score or only overall, because the sensory analysis was performed by a panel of trained assessors.

AU: thank you for this important suggestion. We modified it in table 9 and in the text.

References should be described as follows, depending on the type of work:

  • Journal Articles:
    1. Author 1, A.B.; Author 2, C.D. Title of the article. Abbreviated Journal Name YearVolume, page range.

Check all references: Line 455, 482, 504, 522, 529, 533, 535, 539, 541, 556, 571, 594 - no journal name

Meat Sci. instead Meat science

Line 478 - check Chemical–nutritional…

Line 488 exist “beef %J Animal Production Science”

Line 509 exist “Organic %J Annals of Animal Science”

Line 524 exist “life1. Journal…”

AU: thank you for your suggestions. Reference list was revised.

Reviewer 2 Report

In this study, the authors aimed to evaluate effect of dietary supplementation of the PTHL on oxidative stability, volatile compounds characteristics and sensory attributes of dry aged beef steaks.

Overall, this study is well desinged and the experiments were well conducted, which led to scientific novel findings regarding effect of PTHL supplement. Also, the manuscript, well organized and written, contains useful information for readers.  

Minor comments

line 28: Initials of Aldehydes and Sulphur should not be capital

line 111: fresh may be freshly?

line 312: Initials of Hydrocarbons, Aromatic, Aldehydes should not be capital

Author Response

In this study, the authors aimed to evaluate effect of dietary supplementation of the PTHL on oxidative stability, volatile compounds characteristics and sensory attributes of dry aged beef steaks.

Overall, this study is well desinged and the experiments were well conducted, which led to scientific novel findings regarding effect of PTHL supplement. Also, the manuscript, well organized and written, contains useful information for readers.

Au: Thank you very much for your comments. We really appreciate.

Minor comments

line 28: Initials of Aldehydes and Sulphur should not be capital

 Au: it was revised.

ine 111: fresh may be freshly?

lAu: it was revised.

line 312: Initials of Hydrocarbons, Aromatic, Aldehydes should not be capital

 Au: it was revised. We leave Hydrocarbons because it is the first word of the sentence. Thank you for your suggestion.